# Effect of Acute Ketamine Treatment on Sympathetic Regulation Indexed by Electrodermal Activity in Adolescent Major Depression

**DOI:** 10.3390/ph17030358

**Published:** 2024-03-10

**Authors:** Veronika Kovacova, Andrea Macejova, Ingrid Tonhajzerova, Zuzana Visnovcova, Nikola Ferencova, Zuzana Mlyncekova, Tomas Kukucka, Ivan Farsky, Slavomir Nosal, Igor Ondrejka

**Affiliations:** 1Clinic of Psychiatry, Jessenius Faculty of Medicine in Martin, Comenius University in Bratislava, University Hospital Martin, Kollarova 2, 03601 Martin, Slovakia; kovacova400@uniba.sk (V.K.); macejova5@uniba.sk (A.M.); ingrid.tonhajzerova@uniba.sk (I.T.); mlyncekova3@uniba.sk (Z.M.); kukucka17@uniba.sk (T.K.); 2Department of Physiology, Jessenius Faculty of Medicine in Martin, Comenius University in Bratislava, Mala Hora 4C, 03601 Martin, Slovakia; 3Biomedical Centre Martin, Jessenius Faculty of Medicine in Martin, Comenius University in Bratislava, Mala Hora 4D, 03601 Martin, Slovakia; zuzana.visnovcova@uniba.sk (Z.V.); nikola.ferencova@uniba.sk (N.F.); 4Department of Nursing, Jessenius Faculty of Medicine in Martin, Comenius University in Bratislava, Mala Hora 5, 03601 Martin, Slovakia; ivan.farsky@uniba.sk; 5Clinic of Pediatric Anesthesiology and Intensive Care Medicine, Jessenius Faculty of Medicine in Martin, Comenius University in Bratislava, University Hospital Martin, Kollarova 2, 03601 Martin, Slovakia; slavomir.nosal@uniba.sk

**Keywords:** ketamine, major depressive disorder, severe episode, sympathetic regulation, electrodermal activity, depressive symptomatology, adolescence

## Abstract

Ketamine is a potential rapid-onset antidepressant characterized by sympathomimetic effects. However, the question of ketamine’s use in treating adolescents’ major depressive disorder (MDD) is still discussed. Thus, we aimed to study the acute effect of ketamine infusion treatment on sympathetic regulation using electrodermal activity (EDA) in addition to an assessment of depressive symptomatology in MDD adolescents. Twenty hospitalized adolescent girls with MDD (average age: 15.0 ± 1.46 yrs.) were examined before and two hours after a single intravenous infusion of ketamine. EDA was continuously recorded for 6 min, and depressive symptoms were assessed before and two hours after ketamine administration. The evaluated parameters included skin conductance level (SCL), nonspecific electrodermal responses (NS-SCRs), MADRS (questions no. 1–10, total score), and CDI (items A–E, total score). EDA parameters showed no significant changes after the ketamine treatment, and depressive symptoms were significantly reduced after the ketamine infusion. The analysis revealed a significant negative correlation between index SCL and CDI-A, CDI-E, and the total CDI score and between index NS-SCRs and MADRS no. 4 before the ketamine treatment. In conclusion, ketamine improved depressive symptomatology without a significant effect on EDA, indicating its potential safety and efficiency as an acute antidepressant intervention in adolescent MDD.

## 1. Introduction

Depression is a serious mental disorder in adolescence characterized by a 19% prevalence [1]. Notably, depression is associated with a high risk of suicide, which represents one of the leading causes of death in people aged 15 to 24 [2]. Moreover, the quality of life associated with depressive disorder, particularly during adolescence, is significantly altered, including a reduced ability to manage normal daily activities and decreased performance in the educational and social fields [3]. Thus, the safe and effective treatment of depressive disorder in adolescence is very important.

The first-choice antidepressant treatment for adolescents is therapy via selective serotonin reuptake inhibitors (SSRIs) such as fluoxetine following the current NICE treatment recommendations for an adequate SSRI treatment trial lasting at least 8 weeks at or above the minimally recommended dose [4]. Similarly, Dwyer et al. defined an adequate trial of SSRI treatment for adolescent depression as at least 6–8 weeks of treatment on the minimally recommended FDA dose with appropriate titration up to the maximally recommended tolerated dose [5]. Moreover, an adequate MDD treatment response to first-line interventions is defined by at least a 50% reduction in symptoms [6,7]. However, it has been reported that at least 40% of depressive adolescents do not adequately respond to SSRI treatment [5,6]. Moreover, adolescents not responding to first-line MDD treatments have higher suicide rates, greater impairments to their academic and social skills, and more conflicts with family and peers [8]. Importantly, Dwyer et al. (2020) defined adolescent treatment-resistant depression (TRD) as adolescent MDD in which significant depressive symptoms are exhibited despite the patient receiving an adequate trial of a first-line antidepressant agent (i.e., an SSRI such as fluoxetine, escitalopram, or sertraline) and psychotherapy [5]. However, pharmacotherapy interventions for adolescent TRD are still being extensively discussed. In this context, the discovery of the rapid antidepressant effects of intravenous ketamine [9] brought about an effective treatment for adolescent MDD, particularly in TRD patients. Ketamine is a chiral molecule with two enantiomers, R-ketamine and S-ketamine. The mixture of both enantiomers in equal parts is called racemic ketamine, and this represents the most commonly used form of ketamine in clinical settings. The isolation of enantiomeric S-ketamine (esketamine), which has a four-fold higher affinity to N-methyl-D-aspartate (NMDA) receptors, and its availability as an intranasal spray approved by the FDA and EMA for use in adult TRD, could present a more practical option compared to intravenous racemic ketamine [10,11,12]. However, despite its potential benefit, the side effects of esketamine, such as dissociative symptoms, are of comparable severity to intravenous racemic ketamine [13]. Moreover, a recent systematic review and meta-analysis concluded that intravenous ketamine had better efficiency than intranasal esketamine for MDD treatment [14]. While numerous studies have confirmed the fast-acting antidepressant, anti-suicidal, and anti-hedonic effects of ketamine in adults [15,16,17,18], the safety and efficacy of ketamine administration for adolescent depression are still unclear and understudied [19].

From a pharmacokinetic point of view, ketamine metabolism is primarily linked to enzymatic processes in the liver. Specifically, the enzyme responsible for the formation of the main metabolite, norketamine (a pharmacologically active metabolite with analgesic effects), is cytochrome CYP3A4 [20,21]. In the following metabolic processes, ketamine and norketamine undergo hydroxylation, while their hydroxylated derivatives are conjugated with glucuronic acid in the liver microsomal glucuronosyltransferase system. Glucuronidation increases their solubility, facilitating their excretion in bile and urine [22,23,24]. With regard to adolescence, differences in ketamine pharmacokinetics are assumed to occur primarily at the enzymatic level of the metabolism, which is clinically manifested as the need for a higher dose of ketamine per kilogram to produce an anesthetic effect compared to adults. This necessity of a higher dose of ketamine in adolescence is also caused by developmental differences in cerebral circulation and metabolism, the insufficient myelination of neurons, and age-linked changes in the cardiovascular system, such as a higher cardiac output [25,26]. The elimination half-life of ketamine is relatively short (approximately 3 h), but several studies have pointed to slower elimination times after repeated ketamine administration [27,28]. In adolescence, ketamine elimination is faster compared to in adulthood [29]. Moreover, ketamine clearance is around 79 l/h/70 kg in adults [30], while in the pediatric population, it can reach values of up to 90 l/h/70 kg [31].

In addition, the pharmacodynamics of ketamine are complex, including action on various receptors—NMDA, serotonergic, opioid, dopaminergic, sigma, γ-aminobutyric acid type A (GABA-A), nicotinic, and muscarinic acetylcholine receptors—as well as on sodium, potassium, hyperpolarization-activated cyclic nucleotide (HCN) channels, L-type calcium channels, and monoamine transporters [32]. However, the antidepressant effect of ketamine is currently primarily attributed to its action on the postsynaptic glutamate NMDA receptors of GABA inhibitory interneurons in the form of non-competitive antagonism [33]. The subsequent disinhibition of glutamatergic neurons is manifested by a sudden release of glutamate from presynaptic neurons into the synaptic cleft in the medial prefrontal cortex and hippocampus. Next, the released glutamate molecules bind to α-amino-3-hydroxy-5-methyl-4-isoxazole-propionic acid (AMPA)’s postsynaptic receptors, with their activation leading to the disinhibition of the translation of synaptic proteins and brain-derived neurotrophic factor (BDNF). In general, this cascade of reactions leads to an increase in the number of synaptic signaling proteins and novel functional dendritic synapses, allowing massive synaptogenesis to occur in the prefrontal cortex [34]. Therefore, it is assumed that ketamine can restore the physiological connectivity between the prefrontal cortex and the limbic system, which plays a key role in the emotional regulation/dysregulation associated with depressive disorder [35].

Furthermore, a recent meta-analysis provided evidence for an association between autonomic nervous system dysfunction and mental disorders characterized by emotional dysregulation in youths [36]. Specifically, sympathetic neural mechanisms play a key role in human cardiovascular health and disease, including hypertension and heart failure [37]. In this respect, electrodermal activity (EDA) represents a promising, noninvasive tool for measuring sympathetic regulation, which is closely related to emotional and cognitive states [38]. In other words, EDA reflects changes in the electrical properties of the skin related to the sudomotor activity of the eccrine sweat glands that are regulated via the cholinergic sympathetic nervous system [39,40,41]. The basic principle of EDA evaluation is the measurement of water and electrolyte secretion between the two electrodes placed on the surface of the skin on the phalanges, palms, or feet. As the glands on the palms or soles are also activated in response to emotional stress, EDA can reflect an individual´s level of emotional arousal [42]. With respect to adolescent depression, recent studies have revealed sympathetic hypoactivity, indexed by a reduced EDA, which is associated with a higher risk of potential cardiovascular complications [43,44]. However, studies concerning sympathetic regulation/dysregulation and the acute effect of ketamine treatment are rare. Although previous studies have shown that ketamine administration may be associated with increases in blood pressure and heart rate, indicating a sympathomimetic effect [45,46], there are no studies on the acute effect of ketamine on sympathetic regulation indexed by EDA in adolescent depression.

This study is focused on two goals. The first aim is to study the acute effect of ketamine treatment on sympathetic regulation using EDA in major depressive disorder in adolescence. The second goal is to assess the acute effect of ketamine treatment on depressive symptomatology in severe episodes of adolescent MDD. To the best of our knowledge, this is the first study to analyze the acute effect of ketamine on sympathetic neural activity and depressive symptomatology in major depression and severe episodes in adolescents.

## 2. Results

### 2.1. Characteristics of the Studied Group

The basic descriptive characteristics of the evaluated MDD group are presented in Table 1.

### 2.2. The EDA and Hemodynamic Parameters

No significant changes were found in the SCL and NS_SCRs parameters before and after treatment (*p* = 0.820 and *p* = 0.823, respectively). Similarly, the mean values of the hemodynamic parameters—HR, SBP, and DBP—did not show significant changes before and after treatment (*p* = 0.106, *p* = 0.544, and *p* = 0.776, respectively). All results are summarized in Table 2.

### 2.3. Depressive Symptomatology

The MADRS no. 1, MADRS no. 2, MADRS no. 3, MADRS no. 6, MADRS no. 7, MADRS no. 8, MADRS no. 9, MADRS no. 10, and the MADRS total score indices were significantly higher before treatment compared to after the treatment period (*p* < 0.001, effect size = 1.532; *p* < 0.001, effect size = 1.394; *p* < 0.001, effect size = 1.268; *p* = 0.003, effect size = 0.954; *p* < 0.001, effect size = 1.448; *p* = 0.006, effect size = 0.794; *p* = 0.001, effect size = 0.818; *p* < 0.001, effect size = 1.701; and *p* < 0.001, effect size = 1.574, respectively). No significant changes were found in the remaining parameters. The CDI A, CDI E, and CDI total score indices were significantly higher before treatment compared to after the treatment period (*p* = 0.012, effect size = 0.389; *p* = 0.002, effect size = 0.572; and *p* = 0.018, effect size = 0.361, respectively). No significant changes were observed in the remaining parameters. All results are summarized in Table 2.

### 2.4. Correlation Analysis between EDA and Depressive Symptoms

Before ketamine treatment, correlation analysis revealed significant negative correlations between index SCL and CDI total score, CDI A, and CDI E (r = −0.546, *p* = 0.013; r = −0.527, *p* = 0.017; and r= −0.632, *p* = 0.003, respectively) and a significant negative correlation between index NS_SCRs and MADRS no. 4 (r = −0.588, *p* = 0.006) (Figure 1). No significant correlations were found between the EDA parameters and the remaining depression parameters.

In contrast, after the ketamine treatment, the correlation analysis revealed no significant correlations between the evaluated EDA indices (SCL and NS_SCR) or the CDI and MADRS total scores and individual items.

### 2.5. Correlation Analysis between Hemodynamic Measures and Depressive Symptoms

Correlation analysis revealed no significant correlations between the evaluated hemodynamic measures (HR, SBP, and DBP) or the CDI and MADRS total score and individual items.

## 3. Discussion

This study explored, for the first time, the efficacy and safety of acute intravenous ketamine administration by assessing depressive symptomatology in concert with an evaluation of ketamine’s acute effect on sympathetic regulation using EDA in the little-studied field of adolescent depression. Our results revealed significantly decreased depressive symptoms as evaluated using CDI and MADRS, indicating the rapid efficacy of ketamine administration; this decrease in symptoms was associated with non-significant changes in EDA parameters after ketamine treatment. Moreover, significant negative correlations between EDA parameters and several depressive symptoms before treatment indicate that the severity of depression is negatively associated with sympathetic hypoactivity before the ketamine treatment but not two hours after of the ketamine administration. Several mechanisms explaining this finding are suggested.

Our study revealed an improvement in depressive symptomatology through a significant reduction in the MADRS and CDI total scores two hours after the intravenous administration of ketamine. These findings are in accordance with the few current studies dealing with the antidepressant effect of ketamine on adolescent depression [2,46,47,48]. The complex effect of ketamine on the central nervous system (CNS) is determined via a relatively large number of affected receptors [17,49,50,51,52,53,54]. More specifically, ketamine acts as a noncompetitive antagonist on postsynaptic glutamate NMDA receptors located on GABA inhibitory interneurons, leading to the disinhibition of glutamatergic neurons. Subsequently, a sudden release of glutamate from presynaptic neurons into the synaptic cleft in the medial prefrontal cortex and several subcortical structures related to emotional regulation is manifested [33]. The release of glutamate via these receptors and intracellular signaling pathways ultimately leads to an increase in synaptic signaling proteins and dendritic synapses [34]. In other words, the effect of ketamine may lead to synaptogenesis in the prefrontal cortex and to the restoration of the physiological connection between the prefrontal cortex and the limbic structures of the brain, which are key structures in emotional regulation [34,35]. In contrast, a recent study revealed that ketamine administration led to a decrease in the global brain connectivity of the prefrontal region [55]. However, several studies have also pointed to ketamine’s other antidepressant effects on the serotonergic [56,57], opioid [58], nicotinic [59,60], and sigma receptors [61,62,63], as well as HCN [64,65,66] and potassium channels [67]. Thus, ketamine’s antidepressant effect appears to be the result of complex mechanisms that have previously been extensively discussed [17]. Moreover, a significant improvement was achieved in the items focused on negative mood (CDI A) and negative self-esteem (CDI E) in this study. In the same way, the MADRS evaluation pointed to an improvement in apparent and reported sadness (no. 1 and no. 2), inner tension (no. 3), concentration difficulties (no. 6), lassitude (no. 7), the inability to feel (no. 8), pessimistic thoughts (no. 9), and suicidal thoughts (no. 10). It seems that ketamine administration is characterized by prompt effectiveness predominantly for affective and anhedonic symptoms (sadness, a loss of gratification, and a loss of feelings and affection toward others), and cognitive symptoms (a diminished ability to think and concentrate) without a significant influence on somatic symptoms (related to sleep, fatigue, and appetite). It is particularly important to note that ketamine administration led to a significant reduction in suicidal thoughts, thus significantly reducing the risk of suicide in depressive adolescents. For this reason, ketamine may be a great choice for acute antidepressant treatment for severe depressive disorder at adolescent age with a high suicide risk due to its rapid anti-suicidal effect. In addition, suicidality is correlated with the presence of anhedonia, defined as the inability to experience pleasure [68]. While several studies have reported the anti-anhedonic effect of ketamine in adults, there has been no evidence of its effect on anhedonia in adolescents [69]. Our study revealed a significant reduction in affective and anhedonic symptoms after an intravenous ketamine treatment in adolescent depressive patients. It is believed that this anti-anhedonic effect is due to the downstream regulation of dopaminergic activity via the glutamatergic system [70].

Furthermore, it is well known that depression linked to impaired sympathetic neural regulation represents a risk factor for cardiovascular complications [37,71,72]. Previous studies revealed that ketamine administration is predominantly associated with a sympathomimetic effect on the cardiovascular system, resulting in an increased heart rate or blood pressure [73,74,75,76,77]. Although it has been suggested that ketamine activates the sympathetic nervous system through direct action on the CNS [78,79], the mechanism behind this has not yet been fully elucidated. It is assumed that ketamine inhibits the formation of centrally synthesized nitric oxide through the reduction in NMDA receptor activity [75]. Moreover, other mechanisms of ketamine’s sympathomimetic action include the systemic release of catecholamines, the release of noradrenaline from sympathetic ganglion neurons, the inhibitory action of ketamine on the vagus nerve and centrally located muscarinic receptors, and the inhibition of the reuptake of noradrenaline in peripheral nerve endings, as well as in the myocardium [73,80,81]. Previously, increases in systolic and diastolic blood pressure, as well as heart rate, were observed following the administration of sub-anesthetic doses of ketamine [73,74,75,76,77]. However, recent studies demonstrated minor transient hemodynamic/neural changes (i.e., in blood pressure and heart rate) after ketamine treatment in adolescent depression [2,46]. Our study revealed a discrete, nonsignificant increase in heart rate without significant changes in blood pressure or EDA parameters. Thus, we assume that our findings can point to ketamine’s safety in terms of acute complications (e.g., cardiovascular) associated with sympathetic dysregulation in adolescent MDD. It is noteworthy that, as a measure of neural-mediated influences on the activity of sweat glands reflecting pure sympathetic cholinergic regulation, EDA cannot indicate the overall functioning of the sympathetic regulatory network. Therefore, the analysis of other sympathetically mediated effectors’ parameters, such as heart rate or blood pressure variabilities, may provide independent and distinct information on sympathetic regulation in response to ketamine treatment.

A remarkable result is the significant negative correlation between the SCL index and the CDI total score, CDI A (negative mood), and CDI E (negative self-esteem), associated with a negative correlation between index NS-SCRs and MADRS no. 4 (reduced sleep) before ketamine treatment. These findings point to a negative relationship between the severity of depressive symptoms and EDA parameters before acute ketamine treatment. Our results are in accordance with other studies revealing lower SCL values in subjects with more severe depressive symptoms [82,83,84,85,86,87]. In other words, MDD patients with more pronounced, subjectively interpreted depressive symptoms had lower tonic EDA activity before ketamine treatment that appeared to improve (no significant correlations between EDA and depressive symptoms) after the acute ketamine intervention.

### Limitations of the Study

The limitations of this study include the relatively small number of depressed patients who participated, all of whom were the same gender (female). Therefore, further research is needed to validate our findings with a larger set of depressed patients including male adolescent patients. Our study was further limited by its open-label design. Therefore, a blind study with a placebo is needed to confirm our findings. Notably, the analysis of other parameters reflecting effectors’ sympathetically mediated regulatory mechanisms, such as blood pressure variability, could contribute to a more precise clarification of the relationship between the acute effect of ketamine and sympathetic regulation/dysregulation. Further research in this field is, therefore, needed.

## 4. Materials and Methods

### 4.1. Ethics Statement

This study was approved by the Ethics Committee of the Jessenius Faculty of Medicine in Martin, Comenius University in Bratislava, the Slovak Republic (56/2021), and by the Ethics Committee of University Hospital Martin, the Slovak Republic (143/2021). All procedures in our study were performed in accordance with the ethical standards of the institutional and national research committee and with the 1964 Declaration of Helsinki and its later amendments or comparable ethical standards. All patients’ legal representatives were properly informed about the study protocol and provided written informed consent.

### 4.2. Subjects

Initially, thirty adolescent female patients, aged 12–18 years, who were suffering from MDD and hospitalized at the Clinic of Psychiatry were examined. Severe MDD episodes were diagnosed according to the Diagnostic and Statistical Manual of Mental Disorders, Fifth Edition (DSM-5) [88], and confirmed by two independent specialists—child and adolescent psychiatrists.

The inclusion criteria were the following: adolescent age, the diagnosis of a severe episode of MDD according to the DSM-5 [88] confirmed by two independent specialists—child and adolescent psychiatrists, and an inadequate response to SSRI treatment after 8 weeks of treatment on a sufficient recommended dose [8].

The exclusion criteria were the following: a personal history of psychotic illnesses or a manic episode (assessed by two independent psychiatrists), the abuse of psychoactive substances, weight abnormalities (obesity, underweight, or overweight), or endocrinological, cardiovascular, neurological, and other diseases potentially affecting the activity of the autonomic nervous system. Moreover, all subjects were instructed not to use substances influencing autonomic nervous activity (caffeine and nicotine) for at least 12 h before the ketamine infusion therapy.

According to the strict inclusion and exclusion criteria, the final homogeneous sample consisted of 20 adolescent females (average age: 15.0 years ± 1.46). We included only females in the study to achieve precise homogeneity of the studied group because of the different prevalence of depression (generally twice as common in females compared to males) [89] and sex-related differences in autonomic nervous system development [90], which might have affected the results of the electrodermal activity as an index of sympathetic neural control. The included MDD patients were previously treated with fluoxetine at a dose of 20 mg, and 7 patients had been treated with sertraline at a dose of 50 mg. Over the following four weeks, the dose of fluoxetine was increased to 40 mg and the dose of sertraline to 100 mg. A flow diagram for the study subjects is presented in Figure 2.

### 4.3. Continual Recording of Electrodermal Activity

EDA measurement was performed the morning before the ketamine infusion and two hours after the end of the ketamine infusion in order to meet the strict standard conditions necessary to avoid disturbing factors during EDA recording (e.g., silence and only one examiner); avoiding these factors could not be ensured during the ketamine infusion since the patients’ active cooperation was essential (due to, e.g., necessary communication, acute psychotherapy during infusion therapy, the presence of other necessary medical personnel, etc.). Additionally, according to some studies, ketamine’s immediate adverse effects (such as dissociative symptoms or a quick sympathomimetic effect) disappear within two hours after ketamine administration [91,92]. This was another reason for measuring EDA two hours after the ketamine infusion. At first, the participants were instructed to sit comfortably and rest in a special armchair for 10 min to avoid the potential effects of stress (from the laboratory environment and the presence of the examiner). Consequently, EDA was continuously recorded with a sampling frequency of 256 Hz (required via hardware) (FlexComp Infinity Biofeedback, Thought Technology, Montreal, QC, Canada) [93] and monitored using two dry Ag–AgCl bipolar electrodes placed on the middle phalanges of two fingers on the left (non-dominant) hand [93,94,95]. Before each examination, the EDA electrodes were carefully cleaned with an alcohol wipe. The baseline phase of the study lasted for 6 min.

#### EDA Evaluated Parameters

Raw EDA recordings were carefully checked, and rare artifacts were manually removed for data analysis. Next, the tonic EDA component was extracted using the 10th-order low-pass finite impulse response filter [96]. Furthermore, the index of the tonic level of the skin’s electrical conductivity—the skin conductance level (SCL, microSiemens (μS)) [97]—was evaluated as the average amplitude of the tonic EDA from 5 min of artifact-free recordings. The SCL evaluates quantitative alterations in the cholinergic sympathetic nervous system. The typical physiological values of the SCL depend on the size of the sensors used. For the 10 mm sensors used in this study, the range is from 0 to 30 μS [40]. Furthermore, the nonspecific electrodermal responses (NS-SCRs) indicating momentary arousal were evaluated as the frequency of spontaneous skin conductance responses occurring without external stimuli [38,41]. The threshold for NS-SCRs evaluation was 0.05 μS [41].

### 4.4. Assessment of Depressive Symptomatology

The clinical assessment of the participants, with a focus on depressive symptoms, was carried out using internationally accepted and standardized assessment scales meeting the criteria of validity and reliability. The following scales were used: the Montgomery–Asberg Depression Rating Scale (MADRS) and the self-reported Children’s Depression Inventory (CDI) scale. The participants were scored before receiving the ketamine infusion and two hours after the ketamine infusion [91,92]. The scales were provided and assessed by specialists from the fields of child and adolescent psychiatry.

#### 4.4.1. The Montgomery–Asberg Depression Rating Scale

MADRS is one of the most commonly used depression rating scales with high inter-rater reliability, validity, and sensitivity to change. It consists of 10 items targeting the core symptoms of depressive disorder: apparent sadness (no. 1), reported sadness (no. 2), inner tension (no. 3), reduced sleep (no. 4), reduced appetite (no. 5), difficulties concentrating (no. 6), lassitude (no. 7), the inability to feel (no. 8), pessimistic thoughts (no. 9), and suicidal thoughts (no. 10). Individual items are rated on a scale from 0 to 6, while a rating can lie on the defined scale levels (0, 2, 4, 6) or between them (1, 3, 5) [98]. The total score ranges from 0 to 60, with a cut-off score for severe depression of 35 points [99].

#### 4.4.2. The Children’s Depression Inventory

The CDI is an internationally accepted self-reported scale for depression in the child and adolescent population that meets the criteria of validity and reliability [100]. It consists of 27 multiple-choice items that are quantified using values from 0 to 2 according to severity. Based on the individual items, five basic groups of depression symptoms are evaluated: CDI A (negative mood), CDI B (interpersonal problems), CDI C (ineffectiveness), CDI D (anhedonia), and CDI E (negative self-esteem). The total score ranges from 0 to 54 and, according to the Kovacs, the recommended cut-off score in clinical settings was set at 13 [101].

### 4.5. Study Protocol

All MDD patients underwent basic laboratory examinations, pregnancy testing, and ECG examination to screen for somatic diseases, the presence of which met the exclusion criteria. All patients were instructed not to consume food for at least 6 h or liquids for at least 2 h prior to the infusion administration. As the MDD adolescents did not respond to previous SSRI antidepressants, SSRI treatment was discontinued, and the intravenous ketamine administration started on the 2nd or 3rd day of the patient’s hospitalization. Ketamine at a dose of 0.5 mg/kg was administered intravenously in the form of an infusion for 40 min under the supervision of specialists in the field of anesthesiology and intensive care medicine. A psychotherapeutic (supportive) intervention was conducted with the patient during the administration of the ketamine infusion. Moreover, the patient’s heart rate and blood pressure were measured using an automated oscillometric device (OMRON M6 Comfort, Kyoto, Japan) before and after ketamine administration. During infusion, vital functions (blood pressure and heart rate) and oxygen saturation were checked every 15 min and then every hour for 2 h after the infusion.

### 4.6. Statistical Analysis

The data were explored and analyzed in jamovi version 1.6.9 (Sydney, Australia). The Shapiro–Wilk normality test was used to evaluate data distributions (Gaussian/non-Gaussian). None of the analyzed data were normally distributed. Consequently, the Wilcoxon rank test was used for comparisons before and after treatment with Bonferroni correction to minimize the experimental and family error rate in multiple comparisons [102]. The associations between the MADRS questionnaire indices and SCL and NS-SCRs and between the CDI questionnaire indices and SCL and NS-SCRs were analyzed using Spearman’s rank-order correlation test with Bonferroni correction applied. The effect size, r, was calculated using G*Power 3.1.9.7 (Dusseldorf University, Dusseldorf, Germany) post hoc: the computation achieved power for the Wilcoxon signed-rank test (matched pairs) from the obtained parameters (means, SDs, and correlations). The parameters were expressed as means ± SDs. A value of *p* < 0.05 (two-tailed) was considered statistically significant.

## 5. Conclusions

Our study revealed a significant improvement in depressive symptomatology predominantly in affective, anhedonic, and cognitive symptoms after the administration of one ketamine infusion in patients with adolescent MDD. Further, ketamine significantly reduced suicidal thoughts, which represents an important finding from the perspective of the suicidality decrease in adolescent MDD. Additionally, the EDA parameters showed no significant changes after the ketamine treatment, indicating ketamine’s relative safety in terms of acute sympathetically mediated complications of adolescent major depression.

## Figures and Tables

**Figure 1 pharmaceuticals-17-00358-f001:**
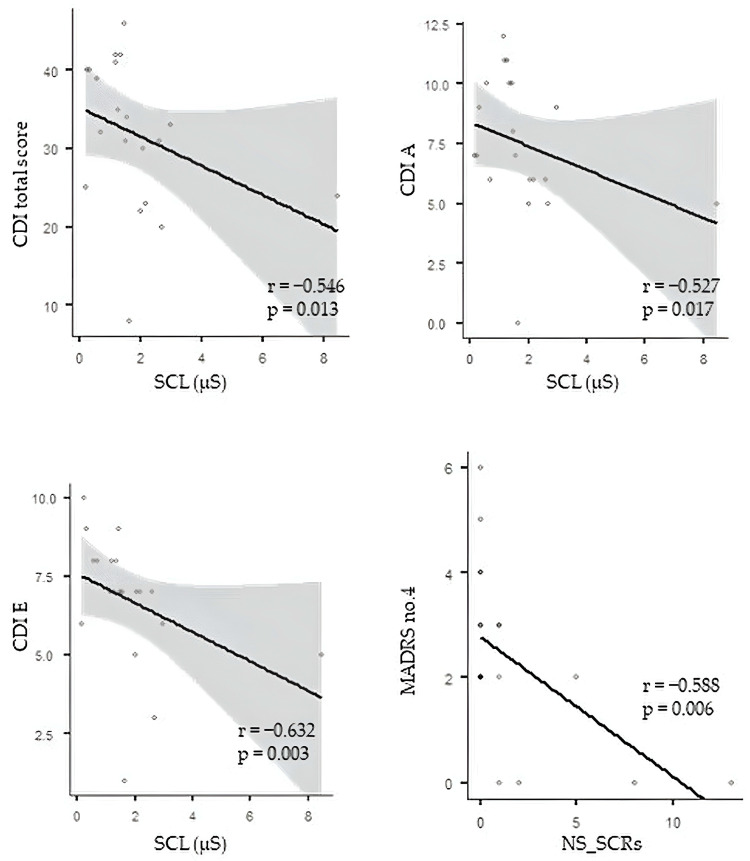
Correlation analysis between EDA and depression measures. CDI—Children´s Depression Inventory; SCL—skin conductance level; MADRS—Montgomery–Asberg Depression Rating Scale; NS_SCRs—nonspecific electrodermal responses. The circles represent individual MDD patients.

**Figure 2 pharmaceuticals-17-00358-f002:**
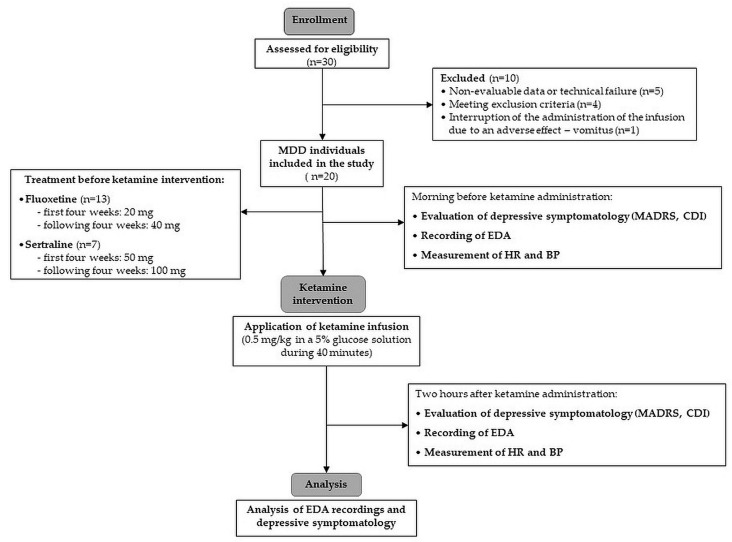
Study flow diagram. MDD—major depressive disorder; MADRS—Montgomery–Asberg Depression Rating Scale; CDI—Children’s Depression Inventory; EDA—electrodermal activity; HR—heart rate; BP—blood pressure.

**Table 1 pharmaceuticals-17-00358-t001:** Characteristics of the studied group.

Number of participants (*n*)	20
Average age (yrs.)	15.0 ± 1.46
Sex	Only female
Mean BMI (kg/m^2^)	20.3 ± 2.68
Current AD treatment	0
Previous AD treatment	Fluoxetine (*n* = 13)
	First four weeks: 20 mg
	Following four weeks: 40 mg
	Sertraline (*n* = 7)
	First four weeks: 50 mg
	Following four weeks: 100 mg

BMI—body mass index; AD—antidepressant. The values are expressed as means ± SDs.

**Table 2 pharmaceuticals-17-00358-t002:** Summary descriptive statistics for all evaluated parameters.

Variable	*n*	Before Ketamine Treatment	Two Hours after the End of Ketamine Treatment	Adjusted *p*-Value
EDA measures				
SCL (μS)	20	1.80 ± 1.77	1.71 ± 1.38	0.820
NS_SCRs	20	1.60 ± 3.36	1.85 ± 3.36	0.823
Hemodynamic measures				
HR (bpm)	20	92.6 ± 7.7	97.0 ± 11.9	0.106
SBP (mmHg)	20	106.6 ± 13.4	108.8 ± 16.6	0.544
DBP (mmHg)	20	62.9 ± 8.5	62.3 ± 9.0	0.776
Depression measures				
MADRS total score	20	32.30 ± 9.21	16.40 ± 10.80	**<0.001**
MADRS no. 1 (apparent sadness)	20	3.95 ± 1.39	1.85 ± 1.35	**<0.001**
MADRS no. 2 (reported sadness)	20	4.35 ± 1.14	2.30 ± 1.66	**<0.001**
MADRS no. 3 (inner tension)	20	3.50 ± 1.05	1.80 ± 1.51	**<0.001**
MADRS no. 4 (reduced sleep)	20	2.35 ± 1.63	1.45 ± 1.57	0.092
MADRS no. 5 (reduced appetite)	20	1.85 ± 2.08	1.20 ± 1.36	0.203
MADRS no. 6 (concentration difficulties)	20	3.10 ± 1.33	1.90 ± 1.17	**0.003**
MADRS no. 7 (lassitude)	20	3.30 ± 1.59	1.15 ± 1.35	**<0.001**
MADRS no. 8 (inability to feel)	20	2.00 ± 1.89	0.70 ± 0.92	**0.006**
MADRS no. 9 (pessimistic thoughts)	20	3.55 ± 1.43	2.35 ± 1.50	**0.001**
MADRS no. 10 (suicidal thoughts)	20	4.35 ± 1.69	1.70 ± 1.38	**<0.001**
CDI total score	20	31.90 ± 9.43	27.6 ± 13.4	**0.018**
CDI A (negative mood)	20	7.50 ± 2.84	6.20 ± 3.68	**0.012**
CDI B (interpersonal problems)	20	3.35 ± 1.31	2.90 ± 1.71	0.083
CDI C (ineffectiveness)	20	5.55 ± 1.96	5.15 ± 2.70	0.413
CDI D (anhedonia)	20	8.75 ± 3.13	7.95 ± 3.78	0.072
CDI E (negative self-esteem)	20	6.75 ± 2.07	5.35 ± 2.70	**0.002**

SCL—skin conductance level; NS_SCRs—frequency of nonspecific skin conductance responses’ occurrence; HR—heart rate; SBP—systolic blood pressure; DBP—diastolic blood pressure; MADRS—Montgomery–Asberg Depression Rating Scale; CDI—Child Depression Inventory. Values are expressed as means ± SDs. The value of *p* < 0.05 (in bold) is considered statistically significant.

## Data Availability

Complete data are available upon reasonable request from the corresponding author.

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
