# Peer review of "Effect of Acute Ketamine Treatment on Sympathetic Regulation Indexed by Electrodermal Activity in Adolescent Major Depression"

_pharmaceuticals, 2024, doi:10.3390/ph17030358_

Round 1
Reviewer 1 Report (Previous Reviewer 1)
Comments and Suggestions for Authors
I thank the authors for thoroughly addressing my comments. With the updates, a few additional items need to be addressed:
-what type of effect size was calculated?
-Table 2 should indicate that the measures were taken 2 hours after the end of the infusion
-Table 2 should also indicate that the p-values listed are adjusted, possibly by using the label "adjusted p-value" instead of "p-value" or having an asterisk next to p-value and explaining this in the legend.
-Please address the fact that one person seems to be driving the significant correlations between SCL and depressive measures. Are the correlations still significant after that person is removed?
Comments on the Quality of English LanguageA native English speaker needs to review the paper for proper word choice and phrasing. For example, the sentence on line 37 of the abstract starting with "Concluding..." should start with "In conclusion..."
Author Response
Please see the attachment.

Reviewer 2 Report (Previous Reviewer 2)
Comments and Suggestions for Authors
My comments have been adequately addressed. Paper can be published as it is.
Author Response
Please see the attachment.

Reviewer 3 Report (Previous Reviewer 3)
Comments and Suggestions for Authors
Dear authors, I completed the review of your article, In my opinion the two main issues are the small small size N=20 : n1=13 and n2=7) which make generalizing the result impossible, also only females were included. Also I have some questions:
Why the study was approved by 2 ethic comittes ?
Also why only female were included in the study : you said higher prevalence 32 against 24 I don’t see a significant difference
Why weight abnormalities were excluded ?
Round 2
Reviewer 3 Report (Previous Reviewer 3)
Comments and Suggestions for Authors
Dear Authors
Thank you for resubmitting your article. Upon reviewing it again, I still find two main issues that, in my opinion, hinder its acceptance. Firstly, the small sample size (N=20; n1=13 and n2=7) limits the generalizability of your results. Additionally, the inclusion of only female participants raises concerns about the generalizability of the findings to broader populations. I appreciate your efforts in revising the manuscript
This manuscript is a resubmission of an earlier submission. The following is a list of the peer review reports and author responses from that submission.
Round 1
Reviewer 1 Report
Comments and Suggestions for Authors
The authors present a study examining the effects of a single ketamine infusion on electrodermal activity parameters, depressive symptoms, and the relationship between these in a cohort of hospitalized adolescent girls. The study's results are relevant to the antidepressant use of ketamine in adolescents, but the following concerns need to be addressed:
-The discussion mentions heart rate and blood pressure as another source of information of sympathetic regulation in ketamine treatment. However, these measures were collected in this study, so it is unclear why they were not analyzed.
-The authors should clarify their definition of "acute complications" given that EDA and other sympathetic parameters would be expected to be elevated during the infusion itself. Why wasn't EDA measured during the infusion? Castillo and colleagues (PMID: 37468572) found that EDA and PR were elevated after IM subanesthetic ketamine.
-Given that the depressive items were analyzed separately for MADRS and CDI, more interpretation should be provided for the items that were significant.
-The rationale for taking measurements two hours after infusion and including only one biological sex should be provided.
-A descriptive statistics table should be added that includes the summary statistics for depression and EDA measures before and after infusion as well as doses of current antidepressants and number of previously antidepressants failed
-The lack of correction for multiple testing should be acknowledged and an explanation provided
-A limitations section should be added to the manuscript that mentions the small sample size, inclusion of a single biological sex, and lack of blinding
Comments on the Quality of English LanguageThe paper requires moderate editing for English grammar.
Reviewer 2 Report
Comments and Suggestions for Authors
Thank you for the possibility to review this paper.
The paper is about the effect of IV ketamine in adolescent MDD, focussing on acute effect of ketamine infusion treatment on sympathetic regulation and MDD symptoms. The paper shows some point of strengths as such as some limitation, particularly regarding the methods section and some analyses.
I would be glad to review the paper after some major revisions. Below are my comments.
Introduction.
Authors should stress the concept that ketamine is mainly an antidepressant used in the resistant forms of depression. They may also add a definition of TRD and some lines about the esketamine enantiomer, currently being approved in many countries as a nasal spray for TRD. Speaking about esketamine and fast acting treatments for TRD, authors may see at: https://pubmed.ncbi.nlm.nih.gov/37331507/
About pharmacological treatment for depression I may suggest this consensus from a nationwide Delphi panel: https://pubmed.ncbi.nlm.nih.gov/37996836/
A more explicative definition for electrodermal activity (EDA) should be added.
Please put the Materials and Methods section before the Results one. This section is to me quite confusing and need some revision.
“Specifically, during the first four weeks, 13 patients were treated with fluoxetine at 246
a dose of 20 mg and 7 patients with sertraline at a dose of 50 mg. Over the following four 247
weeks, the dose of fluoxetine was increased to 40 mg and the dose of sertraline to 100 248
mg.”
This sentence should be put after the statement about the numerosity of the final sample (n = 20), either information about all the 30 subjects initially enrolled should be provided.
No inclusion criteria were provided.
The flow diagram is not clear, e.g., the step “diagnosis of major depressive disorder, severe episode” should be put before the step administration of SSRI medication.
According to which criteria patients were included in ketamine group? Were they unresponsive to SSRI? Or was ketamine administered without waiting for non-response? Were they still on SSRI when receiving ketamine?
Results
Authors should apply a correction in their correlation analysis to adjust p value; e.g., Bonferroni correction for multiple comparisons
Discussion
“Therefore, we suggest that the improvement of depressive symptoms after 185 ketamine acute administration in our MDD patients is the result of a complex effect of 186 ketamine on neuromodulatory systems in the central nervous system. Additionally, it is 187 important to note that just one administration of ketamine infusion can lead to overall 188 reduction of depressive symptoms, and thus to significant improvement of depressive 189 symptomatology including reduction in suicidality.”
The rapid antidepressant effect of ketamine is well known in psychiatry and there is no novelty in authors’ statement. I suggest rather to focus this part of the discussion on ketamine effect in adolescent depression, that can represent a more novel point of this paper (see for example https://pubmed.ncbi.nlm.nih.gov/37732856/)
A limitations section need to be added. Some elements to acknowledge are: only female patients were enrolled, limited sample size..
Reviewer 3 Report
Comments and Suggestions for Authors
Line 46-46 auhtors presented said high prevalence, please present number
Line 48 : One ́s ability to e ‘’ please rephrase
Lines 47-54 please restructure this paragraph in order to have a clear passage between sentences especially the last sentence
Line 56-63 please simplify the sentences
Material and methods
-Why the study received two ethic committee ?
- why the study was concerned on only female ? how the sample was designed ( n=30 ?)
Line 244 SSRI start by writing it in all
Exclusionn criteria : why nicotinims wasn’t excluded from psychoactive substanes ?
Why weight abnormalities were excluded ?
The sample size was 20 where I think insuficcient
Line 260 you stated according to recommdantions of whom ?
Results are not well presented
Discussion the first paragraph just write the main results no need to put 1… 2)…
Abbreviation put in discussion su ch as NMDA GABA were already written in the introduction
Disuccsio needs to be improved
Comments on the Quality of English Languagethere are so many erros that needs to be corrected